# Presence of Ceramidase Activity in Electronegative LDL

**DOI:** 10.3390/ijms24010165

**Published:** 2022-12-22

**Authors:** Núria Puig, Jose Rives, Montserrat Estruch, Ana Aguilera-Simon, Noemi Rotllan, Mercedes Camacho, Núria Colomé, Francesc Canals, José Luis Sánchez-Quesada, Sonia Benitez

**Affiliations:** 1Cardiovascular Biochemistry Group, Research Institute of the Hospital de la Santa Creu i Sant Pau (IIB Sant Pau), 08041 Barcelona, Spain; 2Biochemistry and Molecular Biology Department, Universitat Autònoma de Barcelona, 08193 Cerdanyola, Spain; 3Biotech Research and Innovation Centre, Faculty of Health and Medical Sciences, University of Copenhagen BRIC, DK-1350 Copenhagen, Denmark; 4Vascular Brain Diseases, IIB Sant Pau, 08041 Barcelona, Spain; 5Molecular Basis of Cardiovascular Risk, IIB Sant Pau, 08041 Barcelona, Spain; 6CIBER of Diabetes and Related Metabolic Diseases (CIBERDEM), Instituto de Salud Carlos III, 28029 Madrid, Spain; 7Genomics of Complex Diseases Unit, IIB Sant Pau, 08041 Barcelona, Spain; 8CIBER of Cardiovascular Disease (CIBERCV), Instituto de Salud Carlos III, 28029 Madrid, Spain; 9Proteomics Laboratory, Vall d’Hebron Institute of Oncology (VHIO), 08035 Barcelona, Spain

**Keywords:** electronegative LDL, ceramide, sphingosine, ceramidase, sphingomyelinase

## Abstract

Electronegative low-density lipoprotein (LDL(−)) is a minor modified fraction of human plasma LDL with several atherogenic properties. Among them is increased bioactive lipid mediator content, such as lysophosphatidylcholine (LPC), non-esterified fatty acids (NEFA), ceramide (Cer), and sphingosine (Sph), which are related to the presence of some phospholipolytic activities, including platelet-activating factor acetylhydrolase (PAF-AH), phospholipase C (PLC), and sphingomyelinase (SMase), in LDL(−). However, these enzymes’ activities do not explain the increased Sph content, which typically derives from Cer degradation. In the present study, we analyzed the putative presence of ceramidase (CDase) activity, which could explain the increased Sph content. Thin layer chromatography (TLC) and lipidomic analysis showed that Cer, Sph, and NEFA spontaneously increased in LDL(−) incubated alone at 37 °C, in contrast with native LDL(+). An inhibitor of neutral CDase prevented the formation of Sph and, in turn, increased Cer content in LDL(−). In addition, LDL(−) efficiently degraded fluorescently labeled Cer (NBD-Cer) to form Sph and NEFA. These observations defend the existence of the CDase-like activity’s association with LDL(−). However, neither the proteomic analysis nor the Western blot detected the presence of an enzyme with known CDase activity. Further studies are thus warranted to define the origin of the CDase-like activity detected in LDL(−).

## 1. Introduction

Atherosclerosis involves chronic inflammation in the intima layer of the arterial wall, as triggered by the slow and progressive accumulation of lipids [1]. The main sources of lipids in this process are low-density lipoproteins (LDL), which undergo chemical modification in the subendothelial microenvironment, primarily from lipolytic enzymes and oxidative stress. Modified LDL triggers a strong inflammatory response in monocyte-derived macrophages that infiltrate the arterial wall. 

Electronegative LDL (LDL(−)) is a minor modified form of LDL found in the circulation that accounts for 2–3% of total LDL in healthy subjects. This proportion increases in pathologies associated with cardiovascular disease, such as dyslipemia, diabetes, obesity, metabolic syndrome, or non-alcoholic fatty acid disease [2,3,4], suggesting an active role of LDL(−) in atherogenesis. The atherogenic role of LDL(−) is supported by in vitro studies showing its strong binding with proteoglycans, high susceptibility to aggregation, and induction of a portfolio of inflammatory and apoptotic effects in different cell types from the arterial wall, such as monocytes/macrophages [5,6,7,8], and also lipid accumulation in macrophages [6]. These atherogenic effects seem to be due to an abnormal lipid composition compared to native LDL (LDL(+)). In this context, lipid alterations include increased bioactive lipid mediator content, such as lysophosphatidylcholine (LPC) [9,10], non-esterified fatty acids (NEFA) [9,11], ceramide (Cer), and sphingosine (Sph) [12,13]. Previous studies of ours have described that chemical modifications of total LDL that increase its LPC, NEFA, Cer, and Sph content endow it with inflammatory properties similar to those displayed by LDL(−) [9,11,12,13]. In addition, Cer and Sph have especially been connected to atherosclerosis and CV risk [14,15,16], as well as described to act as crucial signaling molecules that regulate processes such as inflammation, apoptosis, and cell differentiation [17,18,19]. 

The lipid alterations in LDL(−) are, in part, a consequence of at least two phospholipolytic activities present in this particle type. LDL(−) shows increased content and activity of the enzyme platelet-activating factor acetylhydrolase (PAF-AH, also known as lipoprotein-associated phospholipase A_2_, or Lp-PLA_2_) [20], which could be responsible for the particle’s higher LPC content compared to LDL(+)). PAF-AH could be involved in LDL(−)’s increased NEFA content as well, although a fraction of these lipids could also come from its binding to LDL during circulation in situations in which the capacity of albumin for NEFA buffering is exceeded. LDL(−) also shows phospholipase C-like (PLC-like) activity that efficiently hydrolyzes the polar head of choline-containing phospholipids, primarily sphingomyelin (SM) and LPC [21,22,23]. The effect of the PLC-like activity on SM renders phosphorylcholine, a water-soluble molecule that presumably leaves the LDL particle, and Cer, which due to its hydrophobicity is retained in the LDL particles and causes the particle to aggregate [12,24,25]. However, none of these activities explain the increased Sph content in LDL(−).

Besides sphingomyelinase (SMase), a key enzyme controlling sphingolipid metabolism is ceramidase (CDase), which hydrolyzes Cer to yield Sph and NEFA. Therefore, CDases (acid, neutral, and alkaline) are pivotal enzymes that regulate the intracellular homeostasis of Cer/Sph and are critical regulators of signaling pathways within cells [26]. The involvement and potential therapeutic roles of these enzymes in illnesses such as diabetes, obesity, cancer, and neurodegenerative diseases have been extensively reviewed [26,27]. Although there are no data reporting the presence of CDase activity associated with lipoproteins, previous observations have suggested the putative involvement of CDase-like activity in yielding Sph in LDL(−) [13]. Therefore, the aim of the current study was to assess whether Sph content in LDL(−) is a product of intrinsic CDase-like activity.

## 2. Results

### 2.1. Sph and Cer Content in LDL(−) Increase at 37 °C

In a previous study, we reported that the amount of Cer and NEFA in LDL(−), but not LDL(+), was exacerbated when samples were incubated at 37 °C [12]. In the current study, we similarly evaluated the effect of incubation at 37 °C on Sph content. Figure 1 shows that Sph content in LDL(−) strongly increased after 24 h of incubation compared with LDL(+), as measured by thin layer chromatography (TLC) and liquid chromatography-mass spectrometry (LC-MS). In parallel with the Sph increase, LDL(−)’s Cer content also increased during incubation at 37 °C compared with LDL(+), as previously described [12]. 

### 2.2. NEFA Species Increase at 37 °C in LDL(−)

If the increased Sph in LDL(−) is due to CDase activity, the relative NEFA quantity should also increase and have the same order of magnitude as Sph. To test this possibility, we quantified total NEFAs via an enzymocolorimetric assay and determined the different NEFA species with a lipidomic analysis. These data corroborated a previous observation of increased NEFA content in LDL(−) when the particle was incubated at 37 °C [12]. Figure 2a shows the total NEFA in LDL(−) after incubation for 24 h at 37 °C. As expected, the increases in both Sph (Figure 1b,d) and total NEFA (Figure 2a) were similar, approximately two-fold after 24 h. This observation supports the role of CDase activity as being responsible for the generation of both lipids. 

Regarding the NEFA species, their increase in LDL(−) was found to be similar regardless of their classification as saturated (SFA), monounsaturated (MUFA), or polyunsaturated fatty acids (PUFA). Figure 2b shows that the relative quantities of NEFA in LDL(−) versus LDL(+) were similar among the different species (two–three-fold increases), suggesting that the putative CDase-like activity in LDL(−) is able to hydrolyze different ceramide species. 

### 2.3. Effect of CDase Inhibitor on Sph, Cer, and NEFA Content in LDL(−)

Next, we analyzed the effect of the neutral/alkaline CDase inhibitor MAPP (D-erythro-2-(N-Myristoylamino)-1-phenyl-1-propanol). SMase-like and PAF-AH activities in LDL(−) were not altered in the presence of MAPP, meaning this inhibitor did not have an effect on these enzymatic activities (Table 1).

In contrast, Sph and NEFA content in LDL(−) was altered after 24 h of incubation, as measured by TLC (Figure 3a), enzymocolorimetry (Figure 3b), or LC-MS (Figure 3c,d). MAPP partially inhibited the formation of Sph; in turn, LDL(−)’s Cer content was higher in the presence of MAPP. Accordingly, there was a strong decrease in the Sph/Cer ratio in LDL(−) in the presence of MAPP, from 0.78 ± 0.19 to 0.42 ± 0.19 (*p* < 0.05, as measured by TLC/densitometry, *n* = 9). This supports the inhibitory effect of MAPP on CDase-like activity in LDL(−). Lipidomic analysis confirmed the TLC results, revealing a non-significant increase in Cer and a decrease in Sph (Figure 3c). The decreased Sph content in LDL(−) was paralleled by a similarly decreased NEFA content, as measured by LC-MS (Figure 3d) and enzymocolorimetry (Figure 3b). MAPP did not alter the Cer, Sph, or NEFA contents in LDL(+) in any case. 

### 2.4. LDL(−) Is Able to Degrade Cer from an External Source

We also investigated whether the putative CDase-like activity associated with LDL(−) has the ability to degrade Cer molecules external to LDL(−). For this purpose, we used fluorescently labeled Cer (NBD-Cer) as a substrate. The NBD fluorescent probe was bound to Cer’s fatty acid moiety. As a result of the incubation with LDL(−), the NBD-Cer was hydrolyzed and yielded NBD-fatty acid (NBD-FA), whose release was partly inhibited in the presence of MAPP. Figure 4b shows that the NBD-Cer hydrolysis (measured as the formation of NBD-FA) promoted by LDL(−) was partially inhibited in the presence of MAPP.

### 2.5. Proteomic Analysis of LDL(−)

To confirm or reject that a protein with CDase-like activity is bound to LDL(−), we conducted a proteomic analysis. Although previous studies have not shown the presence of these enzymes in LDL(−) [28,29], nor have other proteomic studies conducted on the total LDL (http://www.DavidsonLab.com (accessed on 1 July 2022)) [29,30] reported CDase in these lipoproteins, we repeated these analyses in three of this study’s LDL(−) samples. We identified 22 proteins in LDL(−) and 17 in LDL(+) (Table 2); however, none of these proteins possessed known CDase activity. The mass spectrometry proteomics data have been deposited to the ProteomeXchange Consortium via the PRIDE [1] partner repository, with the dataset identifier PXD036168.

### 2.6. Western Blot of Neutral CDase

To confirm or discard the presence of neutral CDase in LDL(−) observed by proteomic analysis, a Western blot of the LDL(+) and LDL(−) was performed. In contrast with the positive control (protein extract of femoral artery), no evidence of neutral CDase was detected in the LDLs (Appendix A). In order to determine the origin of the CDase-like activity, the possible presence of neutral CDase was also assessed in the water-soluble protein moiety of the LDL fractions after delipidation. Neutral CDase protein was not detected by Western blot in any of the protein fractions (Appendix A).

## 3. Discussion

The present study sheds light on the origin of the sphingolipids found in LDL(−). Compared to native LDL, freshly isolated LDL(−) shows higher Sph and Cer contents, which further increase at 37 °C. Our data support that Cer hydrolysis, triggered by CDase-like activity, is the origin of the increased Sph content in LDL(−). LDL(−)’s ability to hydrolyze Cer has not been attributed to any other lipoprotein. 

Sphingolipids and their metabolizing enzymes are pivotal in the pathophysiology of several diseases, such as type 2 diabetes, Alzheimer’s disease, cancer, and cardiovascular disease, all of these diseases presenting with altered lipid homeostasis. Among the sphingolipids, the bioactive molecules Cer and Sph are key in the signaling of inflammatory pathways [18,19]. In fact, the lipoprotein function and its capacity to modulate inflammation are strongly conditioned by their sphingolipid content [31,32], the amount of which is presumably altered in pathological conditions. A study conducted with diabetic patients showed that their LDL had increased Sph content, which could be related to these patients’ basal inflammatory states and high CV risks [32]. Accordingly, in vitro-enriched LDL with Sph or Cer induced inflammatory cytokine release in monocytes [33,34], making the Sph and Cer content involved in the LDL(−)-induced inflammatory responses in those cells [13]. 

Previous investigations have demonstrated that Cer and NEFA content increase in LDL(−) after incubation at 37 °C for 20 h [12]. These studies have suggested that the increased Cer content originates from the PLC-like activity present in LDL(−), which can degrade SM [21]. Ke et al. [23] confirmed the presence of SMase-like activity uniquely associated with LDL(−). These were not the only studies reporting SMase activity in LDL, since, previously, Holopainen et al. had reported the presence of a similar activity in total LDL [35,36]. Although the origin of such SMase- or PLC-like activity has not been defined, it has been speculated that it could come from structural apoB alterations in LDL(−) [37,38]. Indeed, LDL(−) presents numerous alterations in the conformational structure of apoB compared with LDL(+) [39,40].

Recently, we reported that Sph content is also increased in LDL(−) [13]. The present data confirmed this increase in Sph, which is much enhanced after incubating LDL(−) at 37 °C. In contrast to Cer, which could arise from both de novo synthesis and SM degradation, Sph can only be generated by Cer hydrolysis since there is no de novo Sph synthesis. In this process, cell CDases are essential. CDases are differentially localized within cells and classified depending on their catalytic optimum pH as acid CDase (pH = 4.2–4.3), alkaline CDase (pH = 8.5–9.5), or neutral CDase (pH = 7–9) [27,41]. Both acid and alkaline CDases are intracellular enzymes not found in extracellular compartments. In contrast, neutral CDase is an integral membrane protein that can occasionally be secreted and metabolize Cer in the extracellular milieu [42]. However, several observations from our previous study ruled out the involvement of CDase activity coming from monocytes in mediating LDL(−)-induced inflammatory effects [13]. Therefore, a second possibility is that Cer hydrolysis is driven by the LDL(−) particle itself. Already, our results suggest the involvement of an enzymatic CDase-like activity present in LDL(−) given: (1) the increased Sph content in LDL(−) after incubation at 37 °C in the absence of cells; (2) the equimolar increases in Sph and NEFA at 37 °C; (3) the CDase inhibitor MAPP decreasing the amount of Sph and NEFA in LDL(−); and (4) the degradation of an exogen Cer substrate (NBD-Cer) in the presence of LDL(−), an effect inhibited in the presence of MAPP.

Further evidence of the involvement of CDase-like activity is the NEFA species detected after incubation at 37 °C. At this temperature, the NEFA species that increase in LDL(−) are medium- and long-chain PUFA, such as arachidonic acid. This finding rules out the involvement of PAF-AH in NEFA generation since this enzyme generates short-chain NEFA that would have been generated after the oxidation of the polyunsaturated fatty acid in the sn-2 position of choline-containing phospholipids. Thus, although the origin of the described increased NEFA and Sph content in LDL(−) is not fully determined, Cer hydrolysis is the most plausible source. It is important to point out that the increase in Sph occurs in parallel to the increase in Cer. This observation suggests that SMase-like and CDase-like activities occur at the same time, as a fraction of the formed Cer is degraded to yield Sph. In this way, SMase-like activity seems to be more efficient in degrading SM than CDase-like activity in degrading Cer. 

Our observations indicate that the ability of LDL(−) to degrade Cer is due to neutral, but not acidic, CDase activity, as the experiments were performed at a neutral pH, and MAPP, which inhibits alkaline and neutral CDases, is a poor inhibitor of acid CDase [43]. In addition, C12-NBD-Cer is preferentially hydrolyzed by alkaline and neutral enzymes but not acidic ones [44]. However, neither proteomic analysis nor Western blot found evidence of a specific neutral CDase enzyme’s association with LDL(−). Experiments conducted with delipidated fractions also discarded the implication of water-soluble proteins in the ability of LDL(−) to degrade ceramide. Although our data discarded the presence of the “classical” neutral CDase in LDL(−), the main limitation of the study is that the origin of the CDase-like activity is not clearly defined. As previously suggested for PLC- or SMase-like activities, it could be speculated that some kind of conformational change in apoB promotes CDase-like activity in LDL(−). In that regard, further studies are needed to confirm or rule out the involvement of apoB in the CDase-like activity described in LDL(−). The use of monoclonal antibodies directed at specific epitopes of the apoB sequence [45] could be a useful tool to inhibit CDase-like activity and determine the domain(s) of apoB that could underlie such activity. 

Taken together, the present study highlights that in the absence of cells, LDL(−) shows a hitherto unknown ability to degrade Cer, including exogenous Cer. As far as we know, the unexpected CDase-like activity associated with LDL(−) has not been described for any other lipoprotein. Our findings are relevant since CDase activity is a rate-limiting factor in Sph regulation and, by extension, other bioactive sphingolipid levels, as well as the ensuing biological effects. However, whether this ability to degrade Cer represents a molecular mechanism to counteract or modulate the cytotoxic/inflammatory effects of Cer or not remains unknown. Moreover, since CDase activity encompasses several isoforms whose properties and physiological significance have not been fully elucidated, the comprehensive characterization of this enzymatic activity in LDL(−) and its putative ability to hydrolyze Cer from cell membranes or other lipoprotein particles deserve further investigation.

## 4. Materials and Methods

### 4.1. LDL Isolation

Plasma samples from healthy normolipemic subjects (total cholesterol of <5.2 mmol/L and triglyceride of <1 mmol/L) were obtained and held in EDTA-containing Vacutainer tubes. All subjects gave their written informed consent, and the study was conducted only after we received approval from the Institutional Ethics Committee of the Hospital Sant Pau. All LDL preparations were performed in conditions that prevented oxidation and endotoxin contamination [13]. LDL (1.019–1.050 kg/L) was isolated from plasma via sequential flotation ultracentrifugation at 4 °C, as described in [13]. Total LDL was fractionated into LDL(+) and LDL(−) via preparative anion-exchange chromatography in an ÄKTA-FPLC system (GE Healthcare) [13]. The LDL fractions were then concentrated with Amicon centrifugal filters (Merck Millipore). The LDL(+) and LDL(−) compositions were determined in a Cobas^®^ c501 autoanalyzer, including total cholesterol, triglyceride, apoB (Roche Diagnostics), and NEFA and phospholipid (Wako Chemicals). 

### 4.2. LDL Incubation at 37 °C

LDL(+) and LDL(−) (60 µg apoB) were incubated at 37 °C for up to 24 h in the presence or absence of 10 µM of an inhibitor of neutral/alkaline CDase activity, D-D-erytro-2-(N-myristoylamino)-1-phenyl-1-propanolerytro-MAPP (MAPP) (Cayman). After incubation, the PLC-like and PAF-AH activities in the lipoproteins were analyzed, as previously described in [20,21]. Total cholesterol, triglyceride, NEFA, apoB, and total phospholipids were quantified in a Cobas 501 autoanalyzer, and other lipid components were evaluated with thin-layer chromatography (TLC) or lipidomic analysis, as indicated below.

### 4.3. TLC Lipid Determination

The total lipids were extracted from LDL (50 µg apoB) according to the Bligh and Dyer method; the lipid extract was then dried under a N2 stream, and TLC was performed as described in [12]. Briefly, the lipid extracts were resuspended in 20 µL chloroform and applied to the TLC silica gel plate. Three sequential mobile phases were used (phase 1: chloroform/methanol/water (65/40/5) to 4.5 cm; phase 2: toluene/diethylether/ethanol (60:40:3) to 13.5 cm; and phase 3: heptane to 17 cm). We used Sph (5 µg) and Cer (2 µg) (Sigma) as the controls. Lipids were detected via phosphomolybdate staining, and the quantification was done in a Chemidoc system (BioRad) [12]. The intensity of spots in the TLC assay was normalized using the free cholesterol spot.

### 4.4. Lipid Determination via Liquid Chromatography-Mass Spectrometry 

We evaluated the sphingolipids and NEFA levels in LDL(+) and LDL(−) via LC-MS using the CIBERDEM-Metabolomics Platform from the Universitat Rovira i Virgili (Reus, Spain). 

#### 4.4.1. Sphingolipid Analysis

The lipid extraction from the LDL samples (32 µg apoB) was performed as follows: The LDL samples were lyophilized and resuspended by vortexing in 220 μL of methanol (MeOH). Then, 440 μL of dichloromethane was added and vortexed. Afterward, 140 μL of ultrapure water was added, vortexed, and kept at room temperature for 20 min. The solution was centrifuged for 10 min at 14,500 rpm (4 °C). Upon completion, 400 μL of the organic phase was collected and evaporated to dryness with N_2_ gas. The samples were reconstituted in 150 μL MeOH:toluene (9:1) and transferred to LC-MS vials. The lipid extracts (5 μL) were injected in a UHPLC system (1290 Agilent) coupled with a triple quadrupole (QqQ) mass spectrometer (6490 Agilent Technologies) operated in positive electrospray ionization (ESI+) mode. The instrument was set to acquire in MRM mode. The lipids were separated using C18-RP chromatography (ACQUITY UPLC BEH 2.1 × 150 mm, 1.7 μm, Waters) at 65 °C at a flow rate of 0.4 mL/min. The solvent system was A = acetonitrile:water (60:40) in 10 mM ammonium formate and B = isopropanol:acetonitrile (90:10) in 10 mM ammonium formate. The gradient elution started at 15% B and increased to 30% from minute 0 to 2, 48% B from minute 2 to 2.5, 82% B from minute 2.5 to 11, and 99% B from minute 11 to 11.5. Quality control samples (QC) consisting of pooled samples were injected before the first study sample and then periodically after four study samples. Cer 14:0, Cer 16:0, Cer 18:0, Cer 20:0, Cer 22:0, Cer 24:0, and Sph 18:1 relative concentrations were evaluated with this method. The total Cer was considered the sum of all evaluated Cer species.

#### 4.4.2. NEFA Analysis

Volatile fatty acid methyl ester derivatives (FAMEs) were obtained next [46,47]. Briefly, 20 μL samples were mixed with an IS solution and MeOH and then incubated at 4 °C for 10 min to extract the fatty acids and aid protein precipitation. After this, a derivatization step was carried out using boron trifluoride 14% in MeOH. Afterward, the obtained FAMEs were extracted with hexane before being injected into a GC-MS system. The FAMEs were separated on a HP-88 (100 m × 250 μm × 0.25 μm) column using a temperature program between 140 and 240 °C at 1 mL/min, with He as a carrier gas. Ionization was carried out via electronic impart (70 eV) and a mass analyzer operating on selected ion monitoring (SIM) mode. 

### 4.5. Hydrolysis of External Cer by LDL(−)

To evaluate the ability of LDL(−) to hydrolyze external Cer, LDL(+) and LDL(−) (75 μg apoB) were incubated with CER labeled in the fatty acid moiety with 12-(N-(7-Nitrobenz-2-Oxa-1,3-Diazol-4-yl)amino)hexanoyl) (NBD-CER; Avanti Polar Lipids). The incubation was performed with the fluorescent substrate at 4 μM for 4, 20, and 48 h at 37 °C in the presence or absence of 10 μM MAPP. Total lipids were then extracted and applied to a TLC plate to detect the fluorescent products in a ChemiDoc system (BioRad), as described in [12]. C12 Cer-labeled (NBD-Cer) and arachidonic acid fatty acid-labeled (NBD-NEFA; both 2 μg) were used as the controls for TLC, which was performed with the same three phases described in 4.3. The fluorescence bands were detected using the Chemidoc system (Biorad).

### 4.6. Proteomic Analysis

The protein contents in the LDL(+) and LDL(−) were evaluated by LC-ESI MS/MS on an Esquire HCT ion trap mass spectrometer (Bruker, Bremen, Germany) coupled with a nanoHPLC system (Ultimate, LcPackings, Netherlands) in the Laboratory of Proteomics at the Vall de Hebron Institute of Research (VHIR, Barcelona, Spain), as described in [28]. The lipoproteins were delipidated and solubilized according to Karlsson et al. [48], with minor modifications. The proteins were then digested with trypsin, and the proteomic analysis was performed by LC-ESI MS/MS using 5 µg of protein [28]. Proteins were identified using Mascot (Matrix Science, London, UK) to search the UniProtSwissProt 57.0 human database.

### 4.7. Western Blot of Neutral CDase

The putative presence of neutral CDase in LDL(−) was assessed by Western blot. LDL subfractions (150 µg apoB) were delipidated according to the Bligh and Dyer method. The upper phase contained low-sized water-soluble proteins (wherein neutral CDase would be included), and the lower phase contained the lipids. The protein moiety of LDL(+) and LDL(−), as well as the intact LDL(+) and LDL(−) particles, were assayed for Western blot analysis. LDL(−), LDL(+) (25 μg protein/well), and a protein extract of a femoral artery (as positive control) were run in 10% SDS-PAGE gels. Electrophoresis was performed at 100 V for 2 h at 4 °C. Proteins were transferred to a PVDF membrane (1 h at 30 V, in a tris-glycine buffer containing 20% methanol and 0.1% SDS) and blocked overnight at 4 °C with blocking buffer (Bio-Rad). Western blot was performed using primary mouse monoclonal antibody IgG anti-nCDase (B9 sc374634, Santa Cruz Biotechnology) and secondary HRP-conjugated anti-mouse IgG (m-IgG2b BP-HRP, Santa Cruz Biotechnology). An expanded explanation of this method is shown in the Appendix A.

### 4.8. Statistical Analyses

Differences between the groups were tested using the Wilcoxon t-test for paired data or the Mann–Whitney U-test for unpaired data. The results were expressed as means ± SDs. A value of *p* < 0.05 was considered statistically significant. Data analysis was performed using GraphPad Prism Software Version 6.01 (GraphPad, San Diego, CA, USA).

## Figures and Tables

**Figure 1 ijms-24-00165-f001:**
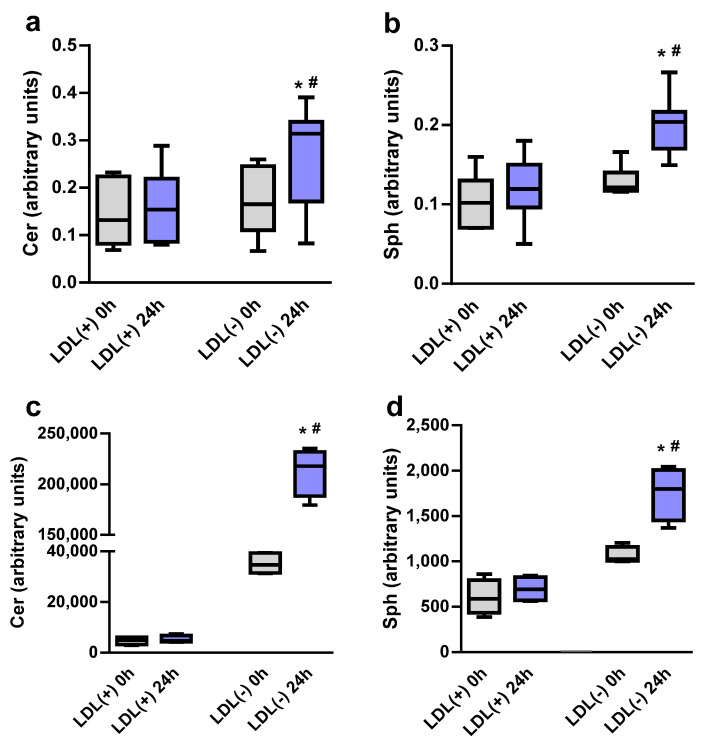
Ceramide (Cer) and sphingosine (Sph) contents in LDL(+) and LDL(−), basal and after 24 h of incubation at 37 °C. (**a**,**b**) Data obtained by TLC (*n* = 9). (**c**,**d**) Data obtained by lipidomic analysis (*n* = 4). * *p* < 0.05 versus LDL(−) at 0 h; # *p* < 0.05 versus LDL(+) at 24 h.

**Figure 2 ijms-24-00165-f002:**
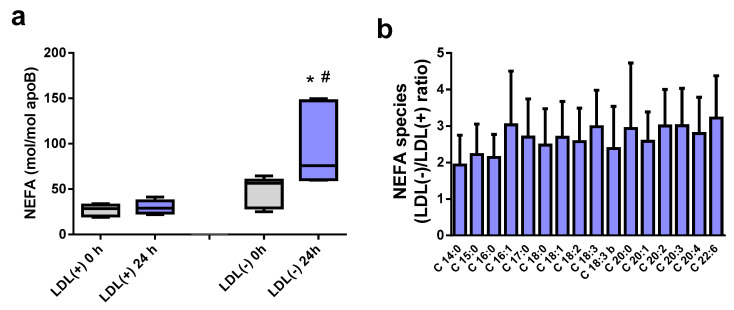
Non-esterified fatty acid (NEFA) content in LDL(+) and LDL(−), basal and after 24 h of incubation at 37 °C. (**a**) Total NEFA content measured via an enzymocolorimetric assay (*n* = 6). (**b**) LDL(−)/LDL(+) ratio of the different NEFAs generated after lipoprotein incubation at 37 °C, as determined by a lipidomic analysis (*n* = 4). * *p* < 0.05 versus LDL(−) at 0 h; # *p* < 0.05 versus LDL(+) at 24 h.

**Figure 3 ijms-24-00165-f003:**
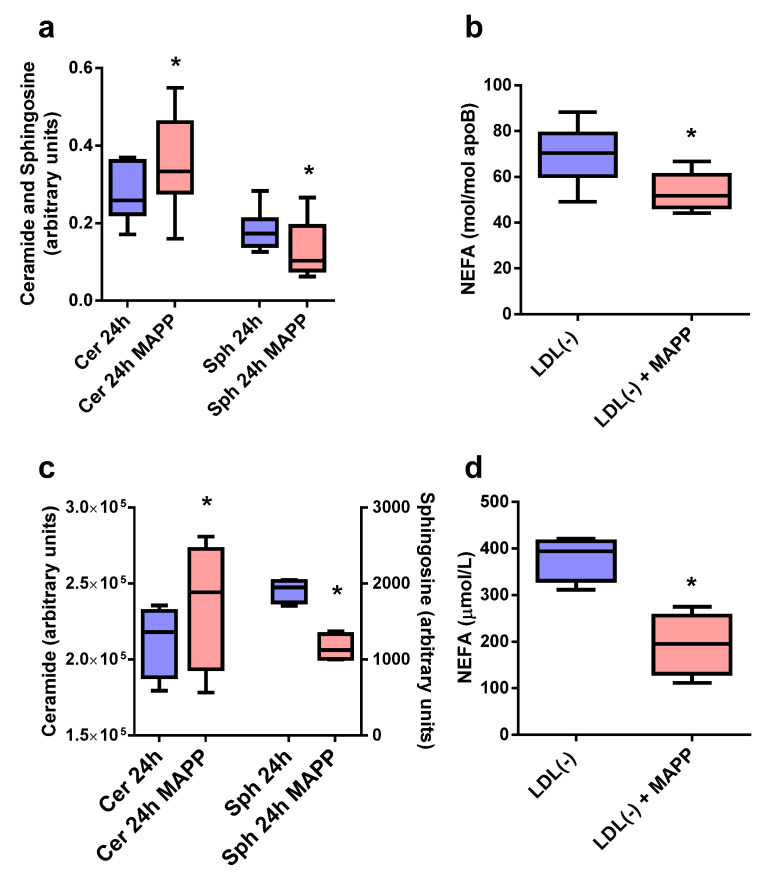
Effect of the ceramidase inhibitor MAPP on the formation of Cer, Sph, and NEFA in LDL(−) after 24 h of incubation at 37 °C. (**a**) Cer and Sph content measured by TLC (*n* = 9). (**b**) NEFA content measured via enzymocolorimetry (*n* = 6). (**c**) Cer and Sph content measured by a lipidomic analysis (*n* = 4). (**d**) NEFA content measured by a lipidomic analysis (*n* = 4). * *p* < 0.05 versus LDL(−) in the absence of MAPP.

**Figure 4 ijms-24-00165-f004:**
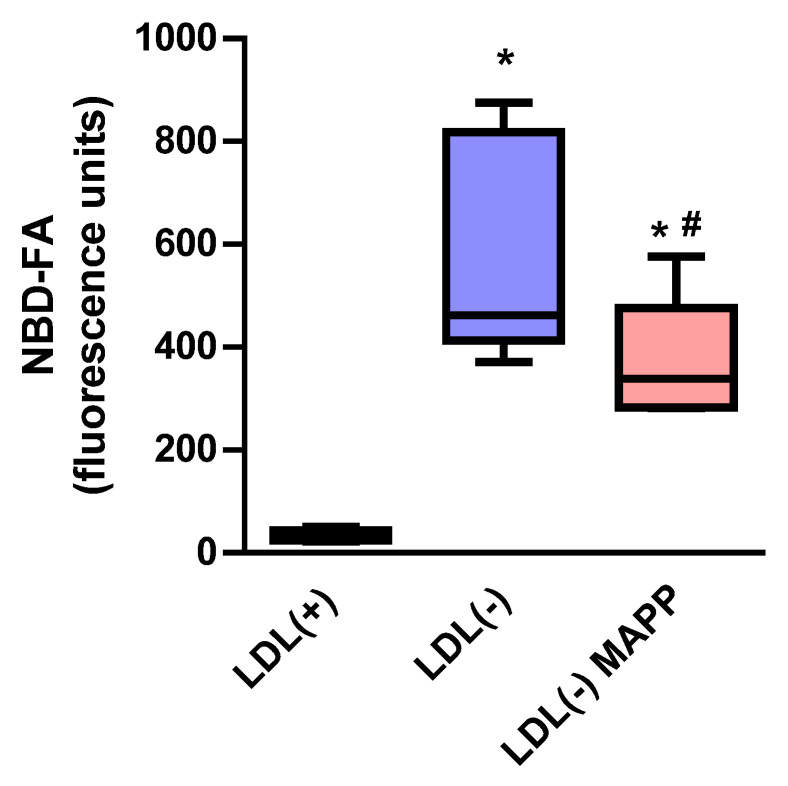
Degradation of NBD-labeled Cer by LDL(−). NBD-Cer was incubated with LDL(+), LDL(−), and LDL(−)+MAPP for 24 h at 37 °C. Degradation of NBD-Cer was measured as the formation of NBD-FA (*n* = 7). * *p* < 0.05 versus LDL(+); # *p* < 0.05 versus LDL(−) alone.

**Table 1 ijms-24-00165-t001:** Effect of D-D-erytro-2-(N-myristoylamino)-1-phenyl-1-propanolerytro-MAPP (MAPP) on platelet-activating factor acetylhydrolase (PAF-AH) and sphingomyelinase (SMase) activities in LDL(−).

	PAF-AH Activity (mmol/min/mL)	SMase Activity (mU/mg apoB)
LDL(−) 24 h	1.25 ± 0.35	205.85 ± 57.77
LDL(−) 24 h plus MAPP	1.29 ± 0.30	215.92 ± 34.48

Data represent the means ± SDs of seven independent experiments.

**Table 2 ijms-24-00165-t002:** Proteomic analysis of LDL subfractions.

		Number of PSMs			% of Total PSMs		Number of PSMs			% of Total PSMs	
Accession	Description	LDL(−)_1	LDL(−)_2	LDL(−)_3	Mean LDL(−)	SD LDL(−)	LDL(+)_1	LDL(+)_2	LDL(+)_3	Mean LDL(+)	SD LDL(+)
ALBU_HUMAN	Albumin OS = Homo sapiens OX = 9606 GN = ALB PE = 1 SV = 2	6	0	3	0.45	0.42	1	19	2	1.05	1.50
APOA_HUMAN	Apolipoprotein(a) OS = Homo sapiens OX = 9606 GN = LPA PE = 1 SV = 1	61	13	63	6.95	4.65	0	0	2	0.09	0.15
APOA1_HUMAN	Apolipoprotein A-I OS = Homo sapiens OX = 9606 GN = APOA1 PE = 1 SV = 1	59	9	32	4.94	3.53	16	14	37	2.92	1.68
APOA2_HUMAN	Apolipoprotein A-II OS = Homo sapiens OX = 9606 GN = APOA2 PE = 1 SV = 1	21	6	7	1.63	1.13	7	5	16	1.22	0.77
APOB_HUMAN	Apolipoprotein B-100 OS = Homo sapiens OX = 9606 GN = APOB PE = 1 SV = 2	460	661	373	71.71	14.23	801	575	645	87.32	5.04
APOC1_HUMAN	Apolipoprotein C-I OS = Homo sapiens OX = 9606 GN = APOC1 PE = 1 SV = 1	4	0	2	0.30	0.28	2	5	4	0.50	0.25
APOC2_HUMAN	Apolipoprotein C-II OS = Homo sapiens OX = 9606 GN = APOC2 PE = 1 SV = 1	7	0	6	0.66	0.57	0	4	1	0.24	0.31
APOC3_HUMAN	Apolipoprotein C-III OS = Homo sapiens OX = 9606 GN = APOC3 PE = 1 SV = 1	8	3	7	0.90	0.43	5	6	4	0.66	0.19
APOC4_HUMAN	Apolipoprotein C-IV OS = Homo sapiens OX = 9606 GN = APOC4 PE = 1 SV = 1	1	0	2	0.16	0.17	0	0	0		
APOD_HUMAN	Apolipoprotein D OS = Homo sapiens OX = 9606 GN = APOD PE = 1 SV = 1	13	11	13	1.82	0.36	7	8	15	1.32	0.59
APOE_HUMAN	Apolipoprotein E OS = Homo sapiens OX = 9606 GN = APOE PE = 1 SV = 1	45	32	46	6.10	1.76	14	28	20	2.78	1.24
APOF_HUMAN	Apolipoprotein F OS = Homo sapiens OX = 9606 GN = APOF PE = 1 SV = 2	3	11	7	1.02	0.54	0	0	3	0.13	0.23
APOL1_HUMAN	Apolipoprotein L1 OS = Homo sapiens OX = 9606 GN = APOL1 PE = 1 SV = 5	2	0	2	0.21	0.18	2	1	3	0.26	0.13
APOM_HUMAN	Apolipoprotein M OS = Homo sapiens OX = 9606 GN = APOM PE = 1 SV = 2	4	0	1	0.24	0.29	1	1	4	0.26	0.23
CLUS_HUMAN	Clusterin OS = Homo sapiens OX = 9606 GN = CLU PE = 1 SV = 1	4	0	8	0.64	0.68	0	0	0		
CO4A_HUMAN	Complement C4-A OS = Homo sapiens OX = 9606 GN = C4A PE = 1 SV = 1	2	2	1	0.24	0.06	0	0	0		
FIBA_HUMAN	Fibrinogen alpha chain OS = Homo sapiens OX = 9606 GN = FGA PE = 1 SV = 2	2	0	0	0.09	0.16	0	4	1	0.24	0.31
PON1_HUMAN	Serum paraoxonase/arylesterase 1 OS = Homo sapiens OX = 9606 GN = PON1 PE = 1 SV = 3	6	0	5	0.56	0.48	0	0	0	0.00	0.00
SAA1_HUMAN	Serum amyloid A-1 protein OS = Homo sapiens OX = 9606 GN = SAA1 PE = 1 SV = 1	3	0	2	0.25	0.22	0	2	0	0.10	0.17
SAA4_HUMAN	Serum amyloid A-4 protein OS = Homo sapiens OX = 9606 GN = SAA4 PE = 1 SV = 1	8	2	10	1.02	0.72	4	11	5	0.91	0.61
SPRL1_HUMAN	SPARC-like protein 1 OS = Homo sapiens OX = 9606 GN = SPARCL1 PE = 1 SV = 2	0	0	1	0.06	0.10	0	0	0		
VTNC_HUMAN	Vitronectin OS = Homo sapiens OX = 9606 GN = VTN PE = 1 SV = 1	0	0	1	0.06	0.10	0	0	0		

Three preparations of LDL(−) and LDL(+) were analyzed by proteomic analysis. The table shows the number of peptide-spectrum matches (PSM) for each protein in each independent sample. Mean ± SD indicate the relative proportion of each protein in LDL(−) and LDL(+). The mass spectrometry proteomics data have been deposited to the ProteomeXchange Consortium via the PRIDE [1] partner repository with the dataset identifier PXD036168.

## Data Availability

The datasets generated and/or analyzed in the present study are available from the corresponding author upon reasonable request. The mass spectrometry proteomics data have been deposited to the ProteomeXchange Consortium via the PRIDE [1] partner repository with the dataset identifier PXD036168.

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
