# Peer review of "Presence of Ceramidase Activity in Electronegative LDL"

_ijms, 2022, doi:10.3390/ijms24010165_

Round 1
Reviewer 1 Report
This study investigated the presence of ceramidase (CDase) and/or CDase activity in the electronegative low-density lipoprotein (LDL(-)) lipoprotein fraction, which could explain the increased sphingosine content in this LDL fraction. Thin layer chromatography and lipidomic analyses showed an increase in ceramide, sphingosine and non-esterified fatty acids in LDL(-) but not in native LDL, when each incubated at 37°C for 24 hours. The use of an inhibitor of neutral CDase resulted in a decrease of sphingosine and an increase in ceramide content in LDL(-). Also, as a result of incubation of LDL(-) with a fluorescently labeled ceramide (NBD-Cer), NBD-Cer was hydrolyzed and generated sphingosine and NBD-fatty acid. The neutral CDase was shown to partly inhibit the release of sphingosine and NBD-fatty acid. However, the authors concluded that proteomic analysis did not detect the presence of an enzyme with known CDase activity.
This study while addressing an important concept in the lipoprotein/sphingolipid field, the study is still in its preliminary stage and could be suitable for a conference abstract or as a reported observation. The current study lacks several possible testable hypothesis, confirmatory methodologies to proteomics, and importantly methods and protocols reported by other laboratories (e.g., delipidation protocols). The authors consistently cited their own work, which may have contributed to the current “uninformative” conclusion. Furthermore, the authors did not propose an alternative or other possible approaches for future experimentation in an effort to decipher the increase in sphingosine in the LDL(-) fraction.
Author Response
Reviewer 1
This study investigated the presence of ceramidase (CDase) and/or CDase activity in the electronegative low-density lipoprotein (LDL(-)) lipoprotein fraction, which could explain the increased sphingosine content in this LDL fraction. Thin layer chromatography and lipidomic analyses showed an increase in ceramide, sphingosine and non-esterified fatty acids in LDL(-) but not in native LDL, when each incubated at 37°C for 24 hours. The use of an inhibitor of neutral CDase resulted in a decrease of sphingosine and an increase in ceramide content in LDL(-). Also, as a result of incubation of LDL(-) with a fluorescently labeled ceramide (NBD-Cer), NBD-Cer was hydrolyzed and generated sphingosine and NBD-fatty acid. The neutral CDase was shown to partly inhibit the release of sphingosine and NBD-fatty acid. However, the authors concluded that proteomic analysis did not detect the presence of an enzyme with known CDase activity.
This study while addressing an important concept in the lipoprotein/sphingolipid field, the study is still in its preliminary stage and could be suitable for a conference abstract or as a reported observation. The current study lacks several possible testable hypothesis, confirmatory methodologies to proteomics, and importantly methods and protocols reported by other laboratories (e.g., delipidation protocols). The authors consistently cited their own work, which may have contributed to the current “uninformative” conclusion. Furthermore, the authors did not propose an alternative or other possible approaches for future experimentation in an effort to decipher the increase in sphingosine in the LDL(-) fraction.
Replies to Reviewer 1
We thank the reviewer for her/his constructive comments. In our opinion, the relevance of the study emerges from the existence of a CDase-like activity in LDL(-), since, as far as we know, the presence of such an activity had not been reported in any lipoprotein. However, we totally agree that the main drawback of the study is that the origin of this CDase-like activity still remains elusive. As a first approach, proteomic analysis seemed to discard the presence of neutral CDase protein, although we are aware that the enzyme could not be detected if the content in LDL was too low. For this reason, we carried out another approach to detect neutral CDase by western blot. In agreement with the proteomic analysis, this enzyme has not been detected in LDL(-) either. This analysis has been included in the supplementary material of the manuscript.
In accordance with the suggestion of the reviewer, we have conducted some experiments using delipidated LDL(-) preparations. For this purpose, LDL fractions were delipidated according to Bligh and Dyer method. The aqueous fraction was collected and concentrated by centrifugal ultrafiltration. This fraction contains all the minor exchangeable proteins in LDL (which should contain neutral CDase if it is present) and was also analyzed by WB; however, as in the case of the whole LDL particle, CDase was not detected in this protein fraction.
In summary, neutral CDase was not detected in the water-soluble protein fraction of LDL(-). We think that these observations discard the presence of the “classical” neutral CDase protein in LDL(-). In this sense, the CDase-like activity in LDL(-) could be compared to the lysophospholipase C/SMase-like activity ascribed to LDL(-). Although this lysoPLC/SMase-like activity has been described by several independent laboratories (Holopainen et al ref #37, Ke et al ref #29, Bancells et al refs #21,22), its origin has not been clearly defined yet. In our opinion, the difficulty lies mainly in the fact that these activities could be generated in apoB itself by changes in its conformation, and given the hugely complex structure of this protein, its detailed study is challenging. The rest of the analyses carried out by our laboratory or other research groups, either in LDL(-) or in total LDL, have failed to find a specific enzyme with lysoPLC/SMase-like activity.
We are planning further experiments in order to figure out the origin of both CDase-like and lysoPLC/SMase-like enzymatic activities in LDL(-). These experiments will assess the role of apoB-100 in such activities. We hypothesize that CDase-like and lysoPLC/SMase-like activities associated with LDL(-) may be generated by conformational changes in apoB-100. In this regard, studies based on apoB sequence by Holopainen et al (ref #37), and Ke et al (ref #29) predicted that residues in the α2 domain of apoB participate in the SMase-like activity of LDL(-). On the other hand, our group, by using an immunochemical approach, reported that the conformational differences of apoB-100 between native LDL and LDL(-) involves both the N-terminal and C-terminal extremes. Hence, we aim to evaluate CDase-like and lysoPLC/SMase-like activities in LDL(-) after blocking apoB epitopes along the protein sequence by using specific antibodies, an experimental approach similar to that previously used in Bancells et al J Biol Chem 2011 (ref #46).
Therefore, our future investigations will address these points, which, in our opinion, are beyond the scope of the present manuscript. We have few days to answer the questions arisen by the reviewers, a time clearly insufficient to conduct the experiments with monoclonal antibodies, which will take at least several months (or even years if new monoclonal antibodies had to be generated). In any case, we are grateful for the interesting comments of the reviewer. In this regard, possible future strategies and limitations of the study are commented in the discussion of the modified version of the manuscript.
Reviewer 2 Report
This manuscript by Nuria Puig et al entitled "Presence of ceramidase activity in electronegative LDL" described the interesting findings about the origin of sphingolipids found in LDL(-) and ceramide hydrolysis, triggered by Ceramidase-like activity, is the origin of the increased Sph content in LDL(-). This distinguishes the atherogenic properties between LDL (-) vs LDL (+), and the authors demonstrated the bench evidence with not only elaborated but relevant data for this important phenomenon. Overall the manuscript is well-prepared and should be accepted in the present form since it achieved the standard of publication in the IJMS. Congratulation!
Author Response
Reviewer 2
This manuscript by Nuria Puig et al entitled "Presence of ceramidase activity in electronegative LDL" described the interesting findings about the origin of sphingolipids found in LDL(-) and ceramide hydrolysis, triggered by Ceramidase-like activity, is the origin of the increased Sph content in LDL(-). This distinguishes the atherogenic properties between LDL (-) vs LDL (+), and the authors demonstrated the bench evidence with not only elaborated but relevant data for this important phenomenon. Overall the manuscript is well-prepared and should be accepted in the present form since it achieved the standard of publication in the IJMS. Congratulation!
Replies to Reviewer 2
We are very grateful for the reviewer's kind comments.
Reviewer 3 Report
This study demonstrates that electronegative LDL (LDL-) has intrinsic ceramidase activity. This was demonstrated by increased levels of sphingosine and NEFA content of LDL(-) upon incubation at 37C for 24 hours and by the increased degradation of exogenously added ceramide, both of which were partially inhibited by a neutral ceramidase inhibitor. The increased ceramide content induced by the incubation likely provided increased substrate for this activity. However, the authors have already published data related to the lipid changes. The impact of the manuscript is also reduced due to the fact that it provides little insight into the molecular mechanism for the changes. The changes in lipid content have already been published by the authors. The more significant issues is that the proteomic analysis in the current manuscript and analyses by other groups have failed to detect ceramidases on LDL(-) and no studies were performed to establish the source of this activity. In addition, the inhibitors only partially attenuated the lipid changes induced by 37C incubation and the degradation of exogenous ceramide suggesting that there may be multiples sources mediating the lipid changes in the LDL(-) and there are concerns about relying solely on inhibitor studies due to potential off-target effects.
Author Response
Reviewer 3
This study demonstrates that electronegative LDL (LDL-) has intrinsic ceramidase activity. This was demonstrated by increased levels of sphingosine and NEFA content of LDL(-) upon incubation at 37C for 24 hours and by the increased degradation of exogenously added ceramide, both of which were partially inhibited by a neutral ceramidase inhibitor. The increased ceramide content induced by the incubation likely provided increased substrate for this activity. However, the authors have already published data related to the lipid changes. The impact of the manuscript is also reduced due to the fact that it provides little insight into the molecular mechanism for the changes. The changes in lipid content have already been published by the authors. The more significant issues is that the proteomic analysis in the current manuscript and analyses by other groups have failed to detect ceramidases on LDL(-) and no studies were performed to establish the source of this activity. In addition, the inhibitors only partially attenuated the lipid changes induced by 37C incubation and the degradation of exogenous ceramide suggesting that there may be multiples sources mediating the lipid changes in the LDL(-) and there are concerns about relying solely on inhibitor studies due to potential off-target effects.
Replies to Reviewer 3
We thank the reviewer for her/his constructive comments. We agree that we have previously reported lipid changes in LDL(-) similar to that described in the present study (Puig et al ref #13). In this line, the present study is a continuation of those investigations. From our point of view, the major novelty of the manuscript is to establish the existence of a CDase-like activity in LDL(-) that is involved in the increased NEFA and Sph content. In that regard, although the alterations in the lipid content of LDL(-) had already been described, they were not attributed to an intrinsic CDase-like activity. As far as we know, such an activity had not been reported in any lipoprotein and this represents a major paradigm shift regarding the properties of LDL.
We also agree on the lack of results discerning the source of this CDase-like activity. Our first objective was to rule out or confirm the presence of a protein in LDL(-) with CDase activity. Proteomic analysis was a first approach, but the negative results does not totally discard the presence of a protein with neutral CDase activity. In the modified version of the manuscript western blot analyzing the putative presence of neutral CDase (which was negative) has been included. In addition, we also analyzed the protein fraction of LDL(-) after delipidation using the Bligh and Dyer method (the upper aqueous phase containing minor soluble proteins, and the lower phase containing lipids). WB analysis also failed to find evidence for the presence of neutral CDase in the protein fraction of LDL(-). Those results are shown in the supplementary material of the manuscript.
Regarding the effect of the inhibitor MAPP, we agree that the CDase-like activity is only partially inhibited by MAPP. However, it must be taken into account that MAPP is a specific inhibitor of the neutral CDase, but it may not inhibit the CDase-like that we describe in this work. By using various concentrations of MAPP, we observed that the formation of Sph was not inhibited more than 30-40%, even by using concentrations up to 10 µM. Considering that the IC50 of MAPP is 5 µM (J Biol Chem. 1996;271(21):12646-54), we think that this lack of complete inhibition at 10 µM supports that the CDase-like activity that we describe in LDL(-) is distinct of “classical” neutral CDase. Otherwise, the reviewer suggests that other sources could mediate the lipid changes that converge with CDase-like activity. As discussed in the manuscript, the increase in NEFA content may also be the result of oxidative processes or of PAF-AH activity. The increase in Cer and Sph content in LDL(-) is likely yielded by the joint action of SMase-like and CDase-like activities associated with LDL(-) by degradation of SM and Cer. On the other hand, in physiological conditions, cells could also be a source of Cer and Sph, which could be transferred to LDL(-). However, in our study (and in our previous study Bancells 2008 for SMase-like, #21) there is no presence of cells contributing to the observed increase in these lipid products at 37ºC.
We think that changes in apoB conformation, perhaps mediated by changes in the lipid environment, may contribute to the appearance of CDase-like activity. This point is the main aim of our on-going investigations in order to study the LDL(-)-associated CDase-like activity in depth. Specifically, we will use a battery of monoclonal antibodies directed against different epitopes of apoB with the aim of blocking CDase-like activity (see the manuscript by Bancells et al J Biol Chem. 2011 Jan 14;286(2):1125-33, ref #46). Unfortunately, we have few days to answer the questions arisen by the reviewers. These immunological studies will take a long time before obtaining reliable results, so they will be part of a future article. In fact, these studies will also include the analysis of the LysoPLC-like/SMase-like activity previously reported by our group (Bancells et al references #21 and 22 in the manuscript). Therefore, we consider that they would be outside the scope of this manuscript. We are aware that the main limitation of our study is that the origin of the CDase-like activity has not been defined. This limitation has been included in the discussion of the manuscript. However, the great technical complexity to study how the structural differences of apoB in LDL(+) and LDL(-) may be the origin of this activity preclude a quick answer to this question.
Round 2
Reviewer 1 Report
The study in its reviewed status does not add much to our current knowledge of the possible association of modified lipoprotein particles with sphingolipid metabolizing factors. The manuscript is still in its preliminary stage and could be suitable for a conference abstract or as a reported observation. The current study still lacks several possible testable hypothesis and confirmatory methodologies. Although the authors proposed alternative and other possible approaches for future experimentation deciphering the increase in sphingosine in the LDL(-) fraction, no preliminary data to support any of the proposed approaches were submitted. We agree with the authors that the expected sphingolipid metabolizing activities could be associated with changes in the conformation of apoB, but this is still at the hypothesis stage. Therefore, in this reviewer’s opinion, the present manuscript is not ready to be published.
Reviewer 3 Report
The authors have provided an additional methodology to examine LDL(-) for neutral CDase protein and provided a strategy for further studies to identify the CDase activity in apoB.